# ScRNA-seq expression of *IFI27* and *APOC2* identifies four alveolar macrophage superclusters in healthy BALF

Xin Li[1] , Fred W Kolling[2], Daniel Aridgides[3], Diane Mellinger[3], Alix Ashare[1,3,]* , Claudia V Jakubzick[1,]*

Alveolar macrophages (AMs) reside on the luminal surface of the airways and alveoli, ensuring proper gas exchange by ingesting cellular debris and pathogens, and regulating inflammatory responses. Therefore, understanding the heterogeneity and diverse roles played by AMs, interstitial macrophages, and recruited monocytes is critical for treating airway diseases. We performed single-cell RNA sequencing on 113,213 bronchoalveolar lavage cells from four healthy and three uninflamed cystic fibrosis subjects and identified two MARCKS⁺LGMN⁺IMs, FOLR2⁺SELENOP⁺ and SPP1⁺PLA2G7⁺ IMs, monocyte subtypes, DC1, DC2, migDCs, plasmacytoid DCs, lymphocytes, epithelial cells, and four AM superclusters (families) based on the gene expression of *IFI27* and *APOC2*. These four AM families have at least eight distinct functional members (subclusters) named after their differentially expressed gene(s): IGF1, CCL18, CXCL5, cholesterol, chemokine, metallothionein, interferon, and small-cluster AMs. Interestingly, the chemokine cluster further divides with each subcluster selectively expressing a unique combination of chemokines. One of the most striking observations, besides the heterogeneity, is the conservation of AM family members in relatively equal ratio across all AM superclusters and individuals. Transcriptional data and TotalSeq technology were used to investigate cell surface markers that distinguish resident AMs from recruited monocytes. Last, other AM datasets were projected onto our dataset. Similar AM superclusters and functional subclusters were observed, along with a significant increase in chemokine and IFN AM subclusters in individuals infected with COVID-19. Overall, functional specializations of the AM subclusters suggest that there are highly regulated AM niches with defined programming states, highlighting a clear division of labor.

## Introduction

Alveolar macrophages (AMs) are the primary phagocytes within the airspace of the lungs. In response to inflammatory stimuli, AMs secrete cytokines and chemokines to initiate the immune response. Subsequently, AMs and recruited monocytes contribute to the resolution of the inflammatory response to prevent excess damage (Janssen et al, 2011; Mould et al, 2017; Fujii et al, 2021; Hetzel et al, 2021). In addition, they play a major role in maintaining lipid homeostasis in the lung, which is essential for adequate gas exchange and alveolar epithelial integrity.

AMs may hold keys to understanding, preventing, and curing several diseases. They derive from prenatal monocytes, self-renew, and have shown remarkable plasticity by altering their transcriptome in response to the environmental changes (Janssen et al, 2011; Jakubzick et al, 2013; Yona et al, 2013; Misharin et al, 2017; Mould et al, 2019; Aegerter et al, 2020; Arafa et al, 2022). Their transcriptional profile is altered in various inflammatory lung diseases, including asthma (Fricker & Gibson, 2017; Hetzel et al, 2021), chronic obstructive pulmonary disease (O'Beirne et al, 2020), cystic fibrosis (McFarland & Rosenberg, 2009), and cancer (Deriy et al, 2009; Bonfield, 2015; Bruscia & Bonfield, 2016; Casanova-Acebes, 2021; Hey et al, 2021; Li et al, 2021). Therefore, enhancing our current understanding of the normal cellular composition and functional diversity of AM subtypes will help characterize disease-related transcriptional changes more precisely and assess targeting specifically a given AM subtype to restore immune balance.

Single-cell RNA sequencing (scRNA-seq) technology is critical for accomplishing these goals. AMs have traditionally been thought to be a uniform population of cells that can be activated by different disease states (Mould et al, 2019). However, recent scRNA-seq studies have revealed a rich diversity in AMs in bronchoalveolar lavage fluid (BALF) from healthy subjects, with multiple subpopulations that have yet to be characterized in other diseases (Mould et al, 2021).

Our study was designed to investigate the immune cell populations of BALF and potential differences in individuals with mild CF compared with healthy controls (HCs). Interestingly, unbiased analysis of our scRNA-seq dataset identified four AM superclusters based on the gene expression of interferon *α*–inducible protein 27 (*IFI27*) and apolipoprotein C2 (*APOC2*), which has not been previously described. Each supercluster contains four shared

---

[1]Department of Microbiology and Immunology, Dartmouth Geisel School of Medicine, Hanover, NH, USA   [2]Department of Biomedical Data Science, Dartmouth Geisel School of Medicine, Hanover, NH, USA   [3]Department of Medicine, Dartmouth Hitchcock Medical Center, Lebanon, NH, USA

Correspondence: claudia.jakubzick@dartmouth.edu; alix.ashare@hitchcock.org
*Alix Ashare and Claudia V Jakubzick are co-senior authors.

subclusters named after their differentially expressed genes (DEGs): IGF1.AMs, CCL18.AMs, CXCL5.AMs, and Cholesterol.AMs. Beyond the four AM superclusters, we found four additional clusters, which can be categorized into the AM superclusters: CK.AMs, MT.AMs, IFN.AMs, and Cyc.AMs. The most striking AM subcluster was the CK.AMs (chemokine cluster), which further divides into four subclusters with each selectively expressing a distinct combination of chemokines. Hence, there appears to be a division of labor in leukocyte recruitment. Moreover, the chemokine subclusters suggest that there are distinct transcription factors (based on our regulon analysis) that regulate their differentiation and programming. This pattern is analogous to what is found in T-cell biology, where distinct transcription factors regulate the cell differentiation pathway to produce a defined combination of cytokines known as Th1, Th2, Th17, etc. Overall, our dataset opens many new areas for further investigation, including addressing how resident macrophage and monocyte subtypes are altered during disease and what factors regulate their programming.

## Results

### BAL cell populations are conserved across healthy and mild CF subjects

To assess the cellular composition in the non-diseased airspace versus that with CF, four HCs and three subjects with mild CF underwent BAL (donors, Table 1). All three CF subjects had two copies of the F508del *CFTR* mutation. Study subjects were closely age-matched. CF subjects were selected for this study based on well-preserved lung function and general good health. To investigate the impact of aberrant CFTR function on the immune cell composition of the lung, we enrolled CF subjects without significant lung inflammation and who were not on CFTR modulator treatment. Two subjects had been on CFTR modulator treatment previously but stopped more than 12 mo before enrollment because of side effects, whereas the third subject declined treatment with a CFTR modulator because of overall good health. The CF subjects in this study were on a stable medication regimen and had no exacerbations within the past 24 mo (Table 1).

BALF samples were spun, washed, and immediately loaded onto the 10X platform for scRNA-seq. Sequenced cells were processed, normalized, and integrated using the Seurat package (details in the Materials and Methods section). Uniform manifold approximation and projection (UMAP) of all seven samples illustrates a similar cluster distribution across all subjects (Fig 1A). Using a curated gene list, we identified 12 major cell types (curated list of genes: Table S1; complete list of genes: Table S2 and Fig 1B–D) (Gautier et al, 2012; Collin et al, 2013; Gibbings et al, 2015, 2017; Villani et al, 2017; Collin & Bigley, 2018; Brown et al, 2019; Jin et al, 2019; Cai et al, 2020; Leach et al, 2020; Schupp et al, 2020; Mould et al, 2021). As anticipated, the most abundant cell type is the AM, followed by monocytes, cycling myeloid cells, DC2, FOLR2, and SPP1 interstitial macrophages, lymphocytes, epithelial cells, migratory DCs, DC1, plasmacytoid DCs, and cycling lymphoid cells (Fig 1C and D). Next, MetaNeighbor analysis was used to confirm the classification and replicability of the identified clusters

**Table 1.  Demographic characteristics for recruited subjects.**

| Characteristic | CF (n = 3) | HC (n = 4) |
|---|---|---|
| Sex, female % (n) | 33% (1) | 50% (2) |
| Average age, years (SD) | 28 ± 4.6 | 29 ± 6.1 |
| FEV1, per cent predicted (SD) | 89.8 ± 11 | |
| *Pa* colonization % (n) | 67% (2) | |
| *Staph aureus* colonization % (n) | 33% (1) | |
| F508del/F508del % (n) | 100% (3) | |
| CFTR modulator use (within 12 mo) % (n) | 0% (0) | |
| Dornase alpha use % (n) | 100% (3) | |
| Hypertonic saline use % (n) | 67% (2) | |
| Azithromycin use % (n) | 67% (2) | |
| Inhaled antibiotic use % (n) | 0% (0) | |
| Oral/IV antibiotics within 6 mo % (n) | 0% (0) | |
| Pancreatic insufficiency % (n) | 100% (3) | |
| CFRD % (n) | 0% (0) | |
| CF liver disease | 0% (0) | |

within our dataset and with two other recently published datasets, (Crow et al, 2018; Schupp et al, 2020; Mould et al, 2021). Within our dataset, samples derived from HC and CF subjects demonstrated a strong correlation across all cell types (Fig 1E). Similarly, our HC sample macrophages, monocytes, and epithelial cells strongly correlates with healthy cell types from Mould et al (2021), as do our CF sample cell types with CF sputum cell types (Fig 1F and G). All in all, curated genes and robust alignment of our cells with previous studies support the identification of 12 cell types in healthy and CF BALF.

### Four AM superclusters and shared subclusters co-exist in the airspace of the lungs

To define AM heterogeneity, we isolated and re-clustered the AM population. Unbiased clustering defined 8 AM clusters. Some clusters were previously described (Gardai et al, 2005), such as the metallothionein, chemokine, IFN-reacting, and cycling AM clusters (complete list of genes: Table S3 and Fig 2A–D). In addition, we identified four novel AM superclusters based on the differential expression of two genes: *IFI27* and *APOC2* (Fig 2B–E): $IFI27^+APOC^+$ AMs.S1, $IFI27^+APOC^-$ AMs.S2, $IFI27^-APOC^+$ AMs.S3, and $IFI27^-APOC^-$ AMs.S4. Besides *IFI27* and *APOC2*, the four AM superclusters share the same top DEGs *IFI6*, *CTSC*, *RPS4X*, *RBP4*, *FBP1*, and *GSN*, which suggest anti-microbial and metabolic function (i.e., common macrophage phenotypes) (Fig 2D).

Hierarchically, each AM supercluster contains four shared subclusters. The subclusters were named after their top DEG(s): IGF1, CCL18, CXCL5, and cholesterol (based on the expression of cholesterol-biosynthesis-related genes) (Fig 3A–C). AM Supercluster 4 (AMs.S4) contained two additional subclusters, GF and SNHG. Cells in the GF subcluster express the epithelial growth factor *AREG* and *TGFb* superfamily ligands, including, *GDF15*. This is consistent with observations made in mice, where Areg-secreting AMs contribute to tissue restoration after injury (Xu et al, 2016; Minutti et al,

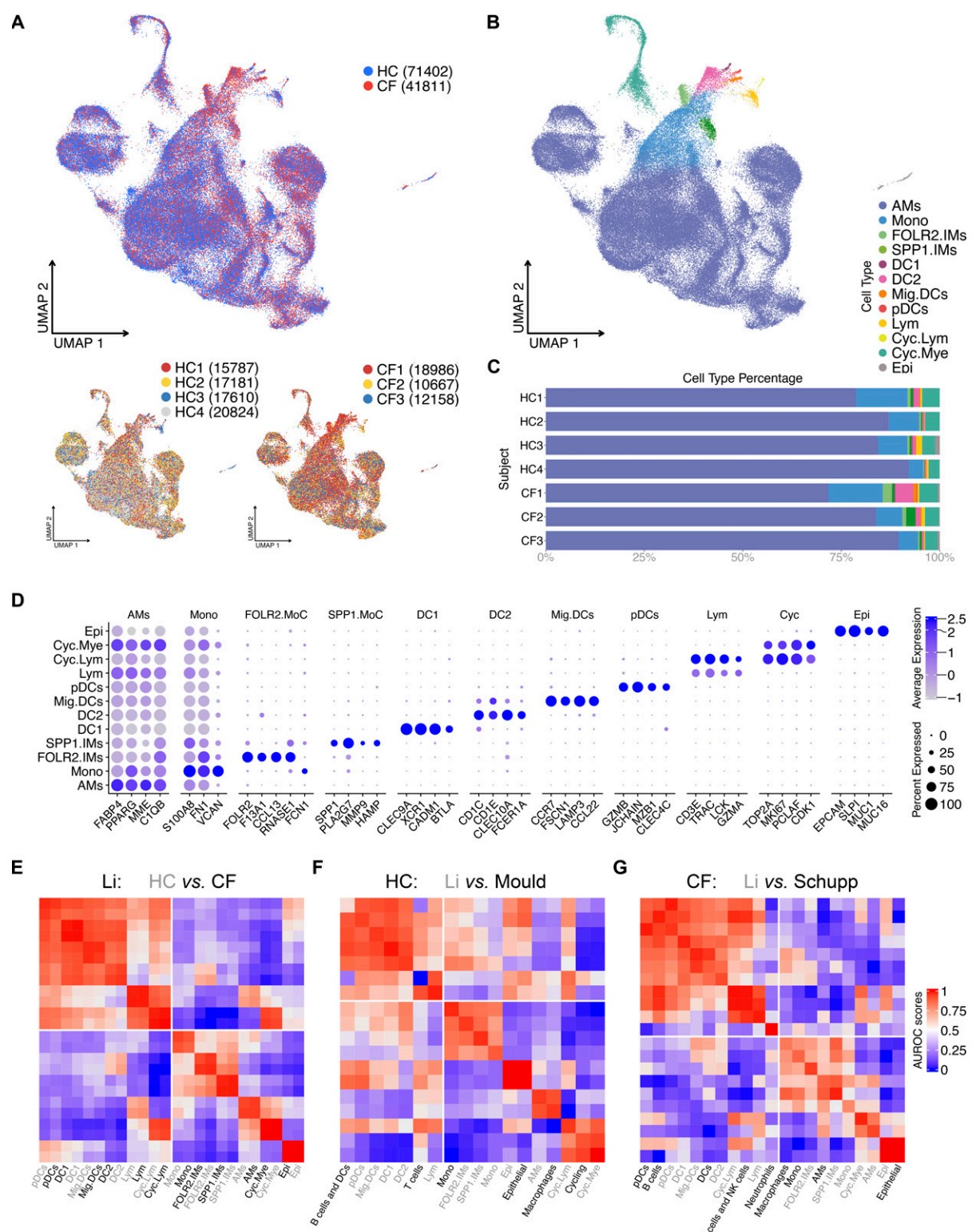

**Figure 1. 12 BAL cell types identified in healthy control (HC) and CF samples.**
**(A)** ScRNA-seq data show integrated UMAP with HC and CF, along with individual sample distribution. **(B)** UMAP demonstrating the 12 major cell types: alveolar macrophages, myeloid and lymphoid cycling cells (Cyc.Mye and Cyc.Lym), FOLR2 and SPP1 interstitial macrophages, monocytes (Mono), DC1, DC2, migratory DC (Mig.DC), plasmacytoid DCs (pDCs), lymphocytes (Lym), and epithelial cells (Epi). **(C)** Percentage bar graph outlines the distribution of individual cell types for each subject. **(D)** Dot plot shows the expression of curated genes in each individual cell type. **(E)** Correlation heat map of cell types against all samples in in-house dataset (Li). The lower quadrant highlights the corresponding cell types. **(F)** Correlation heat map of HC cell types in in-house and Mould dataset. The lower quadrant highlights the corresponding cell types. **(G)** Correlation heat map of HC and CF cell types in in-house and Schupp dataset. The lower quadrant highlights the corresponding cell types.

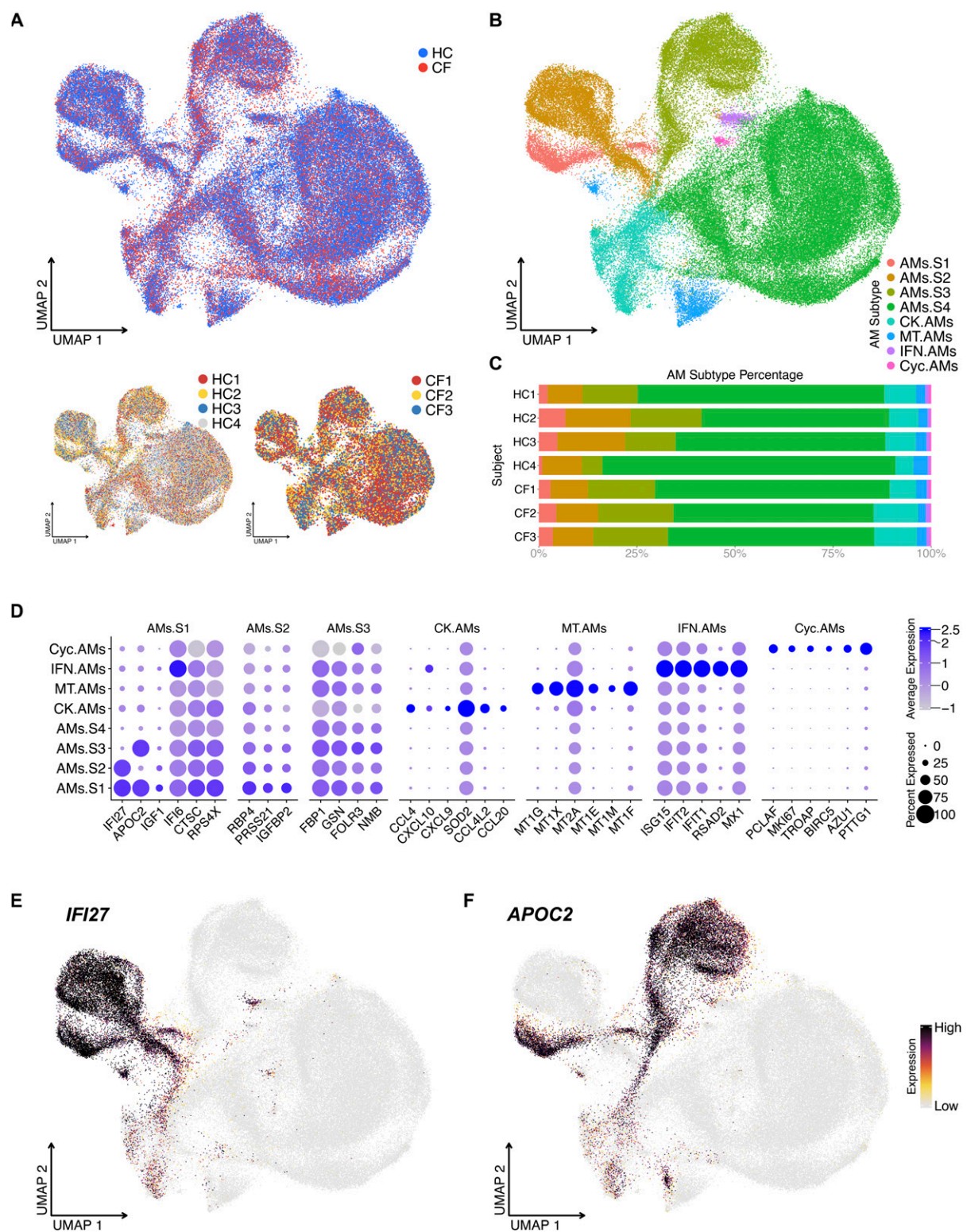

**Figure 2. Unbiased analysis of scRNA-seq data illustrate four alveolar macrophage (AM) superclusters based on *IFI27* and *APOC2* expression.**
**(A)** UMAP of re-clustered AMs shows sample source distribution. **(B)** UMAP shows distinct AM clusters: AM Supercluster 1 (AMs.S1), AMs.S2, AMs.S3 AMs.S4, metallothionein (MT.AMs), chemokine (CK.AMs), interferon (IFN.AMs), and cycling AM (Cyc.AMs) clusters. **(C)** Percentage bar graph outlines the distribution of individual AM clusters for each subject. **(D)** Dot plot shows the expression of the top 6 differentially expressed genes in individual AM clusters, duplicates removed. For example, AMs.S4's top 6 differentially expressed genes are already present in the first three AM superclusters; therefore, AMs.S4 is seen on the Y axis but not the X axis. **(E, F)** Feature plot shows the expression of *IFI27* and *APOC2*, which define the four AM superclusters.

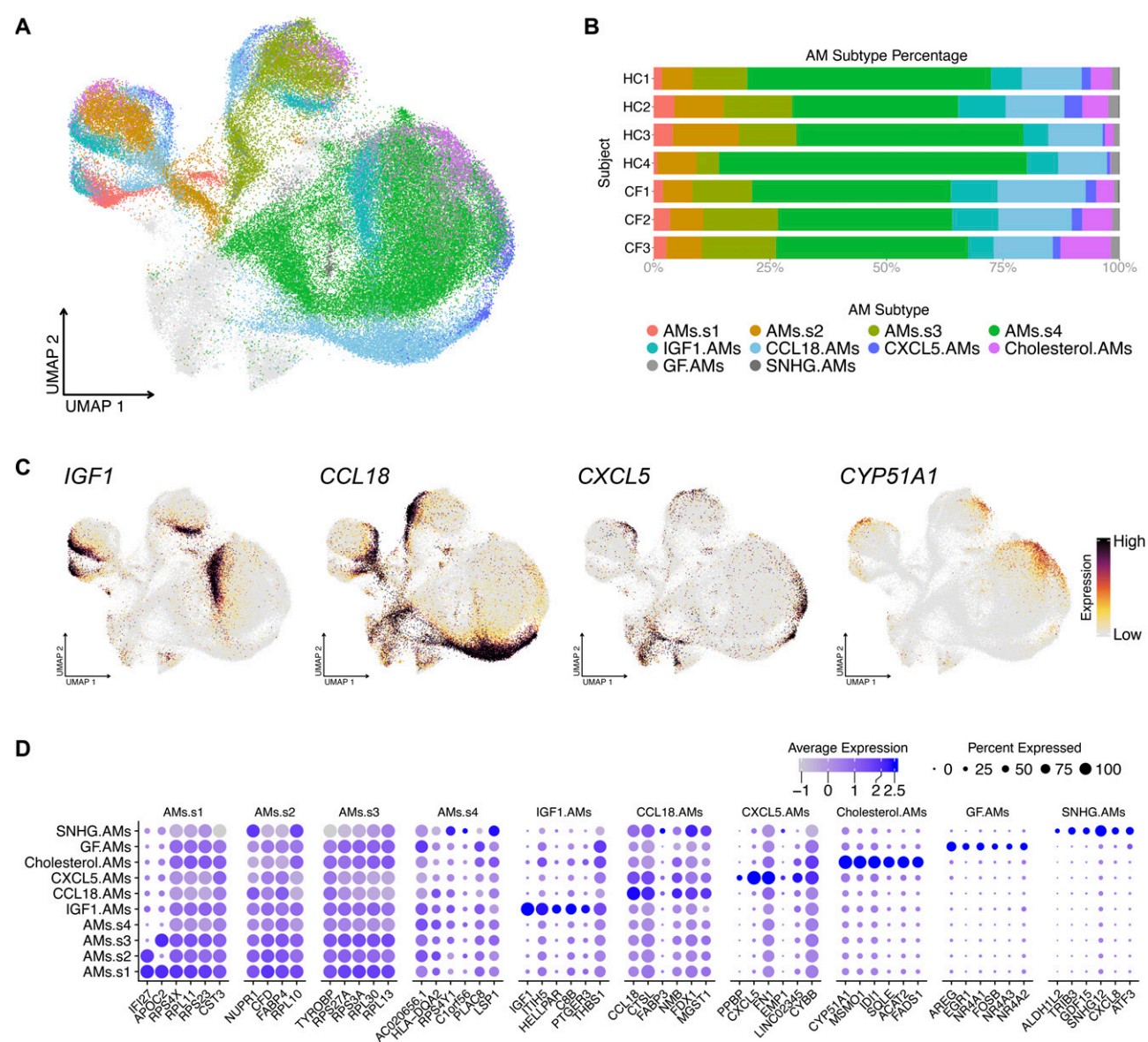

**Figure 3. Alveolar macrophage (AM) superclusters contain four shared subclusters and a small-supercluster.**
**(A)** UMAP illustrates the shared AM subclusters. **(B)** Percentage bar graph outlines the distribution of individual AM subclusters for each subject. **(C)** Featured plots show four representative genes for each shared AM subcluster. **(D)** Dot plot illustrates the expression of the top 6 differentially expressed genes in each individual AM subcluster, duplicates removed.

2019). Whereas cells in SNHG expressed genes encoding small nucleolar RNA host genes (SNHGs) and genes encoding nuclear transcriptional regulators and transcription factors. The last sub-cluster referred to as the small-subcluster within the supercluster, noting a small "s" after AMs (e.g., AMs.s# to differentiate them from their parental superclusters, AMs.S#) had hereditary DEGs, similar to their corresponding superclusters. This small-subcluster was the supercluster minus all other subclusters (complete list of genes: Table S4 and Fig 3D).

In addition to the AM superclusters, four other AM clusters can be defined based on their expression of metallothioneins (MT.AMs), chemokines (CK.AMs), IFN-related genes (IFN.AMs), and cycling genes (CK.AMs) (Fig 2B). The chemokine cluster can be further divided into four subpopulations, which we denote as CK.AMs.c1–4 (Fig 4A). The top 6 DEGs and curated genes illustrate the unique chemokine expression pattern within each subcluster (complete list of genes: Table S4 and Fig 4D). CK.AMs highly express *CCL18*, *CCL23*, and chemokine receptor *CXCR4* in CK.AMs.c1; *CCL3*, *CCL4*, *CXCL8*, and *CCL20* in CK.AMs.c2; *CXCL9*, *CXCL10*, and *CXCL11* (IFN-γ–inducible chemokines) in CK.AMs.c3, and *CXCL3* in CK.AM.c4 (Fig 4B–D). The expression of unique combinations of chemokines in each CK.AMs suggest that different subtypes of AMs are pro-grammed to chemoattract distinct leukocytes into the airspace, depending on the inflammatory stimuli they encounter. Another scRNA-seq study described chemokine-expressing macrophages in human lung tumors: CXCL5-, CXCL12-, and CXCL9/CXCL10/CXCL11-

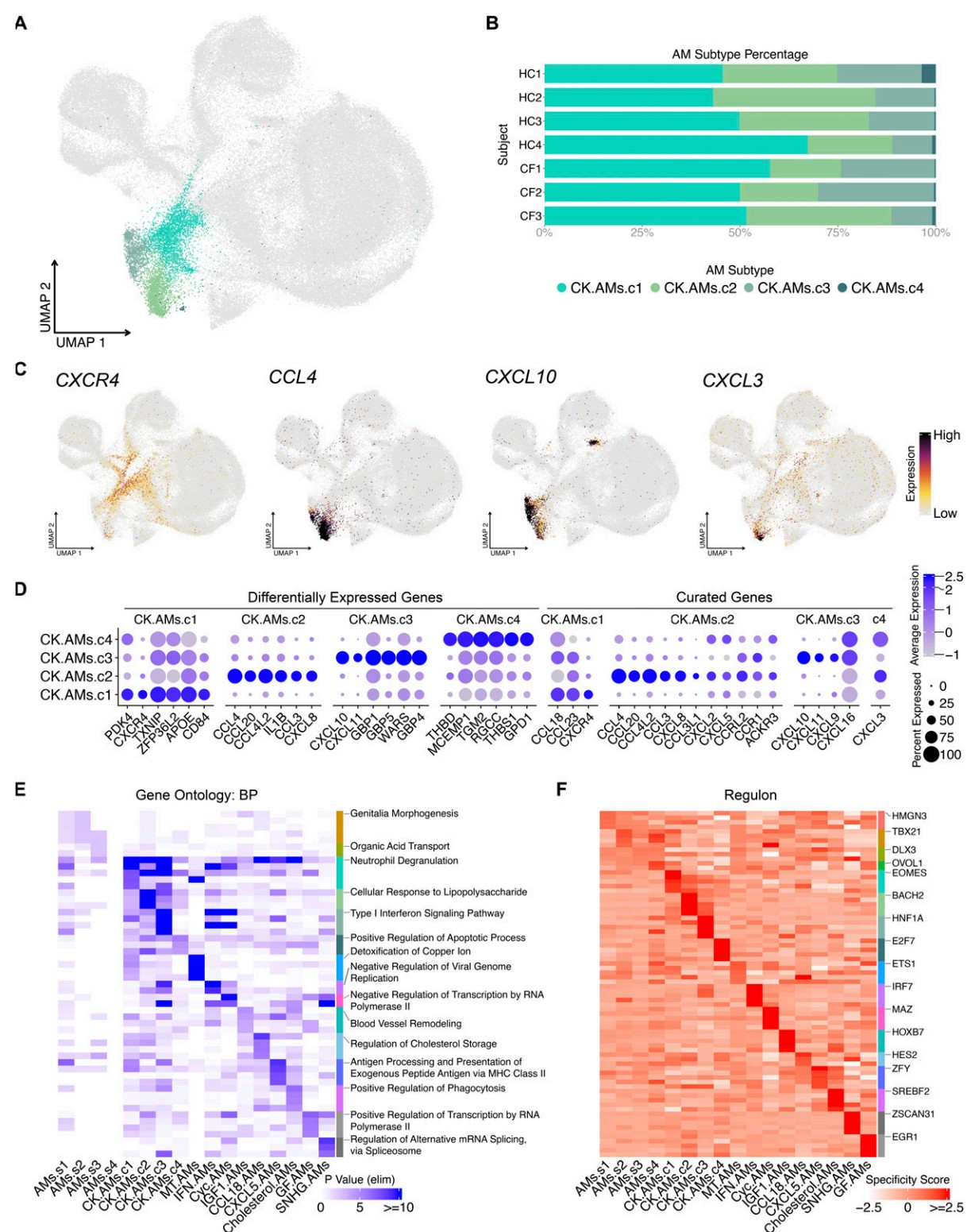

**Figure 4. Chemokine alveolar macrophage (AM) subclusters express unique combinations of chemokines.**
**(A)** UMAP illustrates distinct chemokine-expressing AM subclusters: CK.AMs.c1, CK.AMs.c2, CK.AMs.c3, and CK.AMs.c4. **(B)** Percentage bar graph outlines the distribution of individual chemokine-expressing AM subclusters for each subject. **(C)** Featured plots show four representative gene expression for individual chemokine-expressing AM subcluster. **(D)** Dot plot illustrates the expression of the top 6 differentially expressed genes (duplicates removed) and curated chemokine genes in individual chemokine-expressing AM subcluster. **(E)** Heat map shows the top 5 gene ontology terms (biological process) for each AM subtype. Top 1 term is labeled. **(F)** Heat map shows the top 5 regulons for each AM subtype. Top 1 regulon is labeled.

expressing macrophage subsets (Zilionis et al, 2019). Although these macrophages were not AMs, different macrophage subtypes secrete distinct combinations of chemokines.

Next, given the number of AM subtypes identified, we performed Gene Ontology (GO) enrichment analysis to infer functionality based on their DEGs (Fig 4E). Predictably, the Chemokine AM subclusters contained GO terms regarding cell interaction and response to stimuli, together with other enriched GO terms consistent with the top DEGs in each AM subtype. For instance, the top GO terms for Metallothionein AMs pertain to cellular responses to metal ions (Figs 4E and S1). Likewise, the top four enriched terms for cholesterol AMs contained "*cholesterol biosynthetic process*" and "*regulation of cholesterol biosynthetic process*" (Fig S1).

Last, to understand the transcription factors and co-factors driving each AM subtype, we performed single-cell level regulon analysis using the Single-Cell rEgulatory Network Inference and Clustering (SCENIC) package in R and its Python implementation pySCENIC (Aibar et al, 2017). Co-expressed genes were used to identify and score regulons enriched in each cluster. For instance, the proinflammatory regulator of the cytokine-induced apoptotic pathway, *BACH2*, is the top regulon identified for CK.AMs.c2, whereas the *IRF7* regulon is enriched in IFN-reacting AMs (Figs 4F and S1). This analysis offers yet another vantage point for understanding the distinct AM subtypes and corroborates their defined DEGs and GO terms.

### Like AM chemokine clusters, monocytes also display unique chemokine-expressing clusters

We performed unbiased clustering for monocytes, which outlined eight distinct populations (Fig 5A–C). The largest cluster is denoted as conventional monocytes (Conv.Mono) that express classical myeloid cell markers shared with the other seven clusters (Complete list of genes: Table S5 and Fig 5D). An additional six populations are defined as functional monocytes based on the expression of specific genes, including FCN1.Mono, CCL2.Mono, CXCL5.Mono, CXCL9.Mono, CCL13.Mono, and VEGFA.Mono (Fig 5D). To note, we also identified an intermediate monocyte cluster (Int.Mono) with distinct DEGs that encompasses cells transitioning from Conv.Mono to other functional monocyte phenotypes. Thus, monocytes appear to display a division of labor in orchestrating and recruiting leukocytes into the airspace, similar to what is observed for chemokine-expressing AMs.

### IL10 is selectively secreted by two IM clusters

Other than AMs and monocytes, we observed several unexpected cell type clusters in the BALF, including IMs and DCs. The IMs in our dataset form two distinct clusters with distinct gene expression signatures, which we refer to as FOLR2.IMs and SPP1.IMs (Figs 1D and 5D and E). Genes such as *LGMN*, *MARCKS* (Fig 5E), *TMEM37*, and *MERTK* (data not shown) are specifically expressed by IMs, serving as potential markers to distinguish this population from other macrophage/monocyte cell types. It is worth mentioning that these two IM clusters are the main producer of IL10, suggesting their immunoregulatory role in the airspace.

Next, we performed GO and regulon analysis on the monocyte and IM clusters (Fig S2A and B). The most notable GO term was for FCN1.Mono, whose genes were enriched for bacterial defense

mechanisms suggesting the importance of this monocyte subtype during pulmonary bacterial infections (Fig S2A). The GO analysis also illustrated the exogenous antigen presentation function in FOLR2.IMs and the inflammatory response genes in SPP1.IMs (Fig S2A).

### Pseudotime analysis suggests monocytes differentiate into AMs and IMs

ScRNA-seq captures all cell types at a given time point, producing a series of expression profiles across cells as they transition from one state to another. To define the transitional processes of macrophage development, we performed pseudotime analysis using all the monocytes, AMs, and IMs in our dataset (Trapnell et al, 2014; Qiu et al, 2017a, 2017b; Cao et al, 2019). The trajectory analysis revealed starting points of origin in the monocyte cluster (purple region and white open dots, Fig 6A), revealing two directions of differentiation: one toward IMs and the other toward cycling AMs and AMs.S4 (Fig 6A and B). AMs.S4 appears to have its own starting branch point leading to all other AM clusters (Fig 6A and B). This suggests that AMs.S4 may contribute to all other AM subtypes in the airspace, being the largest supercluster that does not express *IFI27* or *APOC2*.

To pinpoint the genes that change as a function of pseudotime trajectory, DEG analysis was applied. As monocytes differentiate into clusters of AMs and IMs, different genes are up-regulated sequentially, likely as a cause or consequence of changes in their cell state over time (Fig 6C). Genes for cell proliferation and myeloid cell function were among the top DEGs (Fig 6C). For example, *ISG15* is highly expressed during the later progression of pseudotime and is mainly up-regulated in IFN-reacting AMs, whereas another among the top 10 DEGs, *MT2A*, changes along pseudotime and associate with metallothionein AMs. Interestingly, AM superclusters AMs.S1 and AMs.S2 seem to evolve much later than their counterparts AMs.S3 and AMs.S4, even though they share four subclusters and most gene expression, except *IFI27* and *APOC2* (Fig 6C).

Although scRNA-seq has revolutionized our knowledge of macrophage heterogeneity, there are instances where investigators need to isolate bulk populations to perform either functional or morphological analyses. Therefore, flow cytometric antibodies that can be used to sort and enrich bulk AMs from recruited monocytes are needed. Using flow cytometry, we demonstrate that, in addition to high side-scatter for AMs and low side-scatter for monocytes, investigators can use antibodies against CD43 (SPN) and CD169 to define AMs and CD93, CD36, and CD14 to identify monocytes (Fig 6D) (Bharat et al, 2016; Desch et al, 2016; Yu et al, 2016; Gibbings & Jakubzick, 2018; Yu & Tighe, 2018). ScRNA-seq and TotalSeq (for single-cell level protein expression assessment) also support the use of CD169 and CD43 for AMs compared with CD93, CD36, and CD14 for monocytes (Figs 6E and S3). Thus, several forms of technology validate that the higher expression of CD43 and CD169 in BAL cells define AMs over recruited monocytes.

### Bivariate expression of *IFI27* and *APOC2* also exist in other AM subtypes

Although our unbiased AM analysis in Fig 2B resulted in eight clusters, four AM superclusters, and four other clusters (i.e., chemokine, metallothionein, interferon, and cycling AMs), we questioned

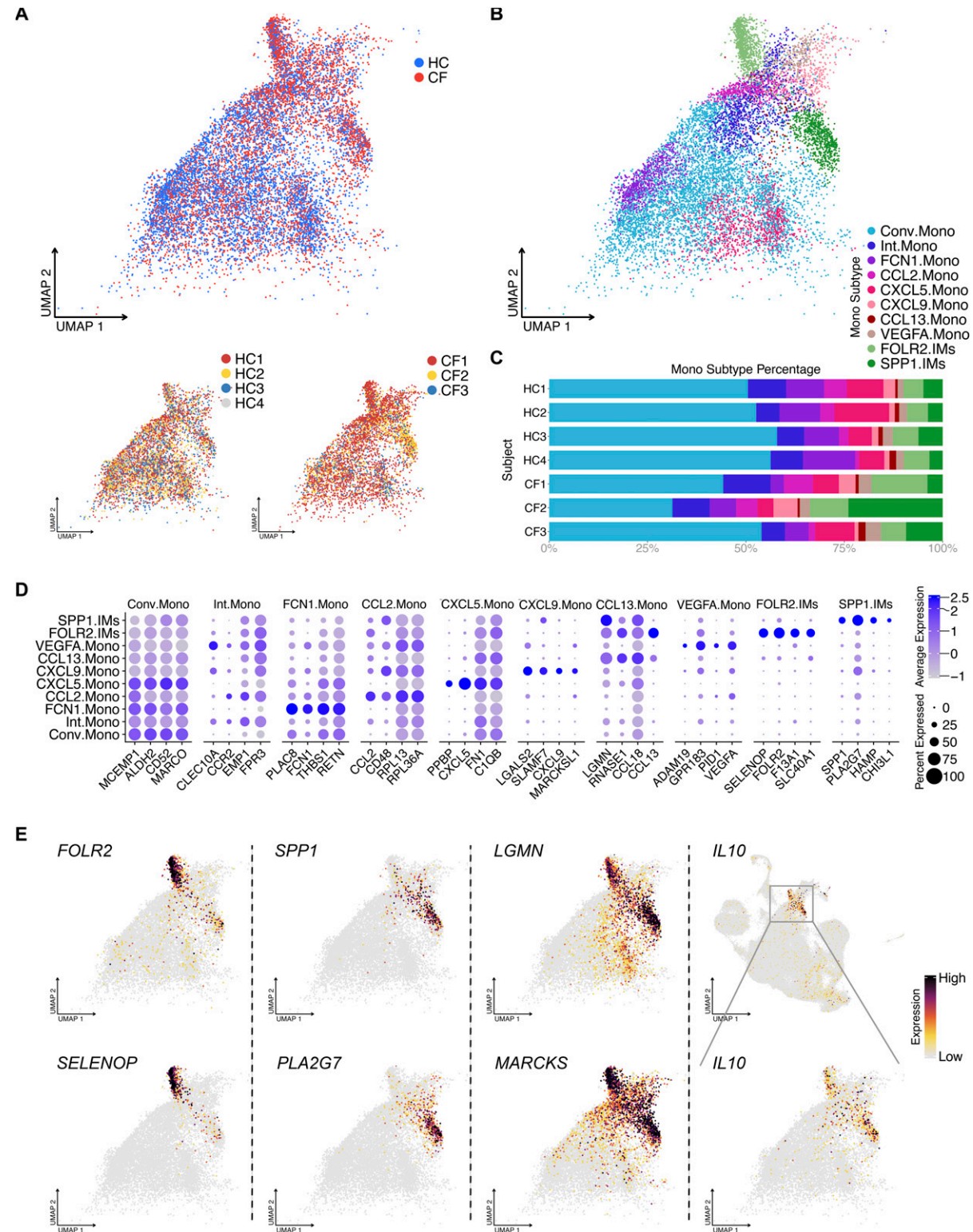

**Figure 5. BAL monocytes and IMs contain heterogeneous clusters.**
**(A)** UMAP of monocytes shows sample source distribution. **(B)** UMAP shows eight distinct monocyte clusters: conventional monocyte cluster (Conv.Mono) and intermediate monocyte cluster (Int.Mono) and others. **(C)** Percentage bar graph outlines the distribution of individual monocyte clusters for each subject. **(D)** Dot plot shows the expression of the top 6 differentially expressed genes in individual monocytes clusters, duplicates removed. **(E)** Featured plots show the expression of different differentially expressed genes of two IM clusters.

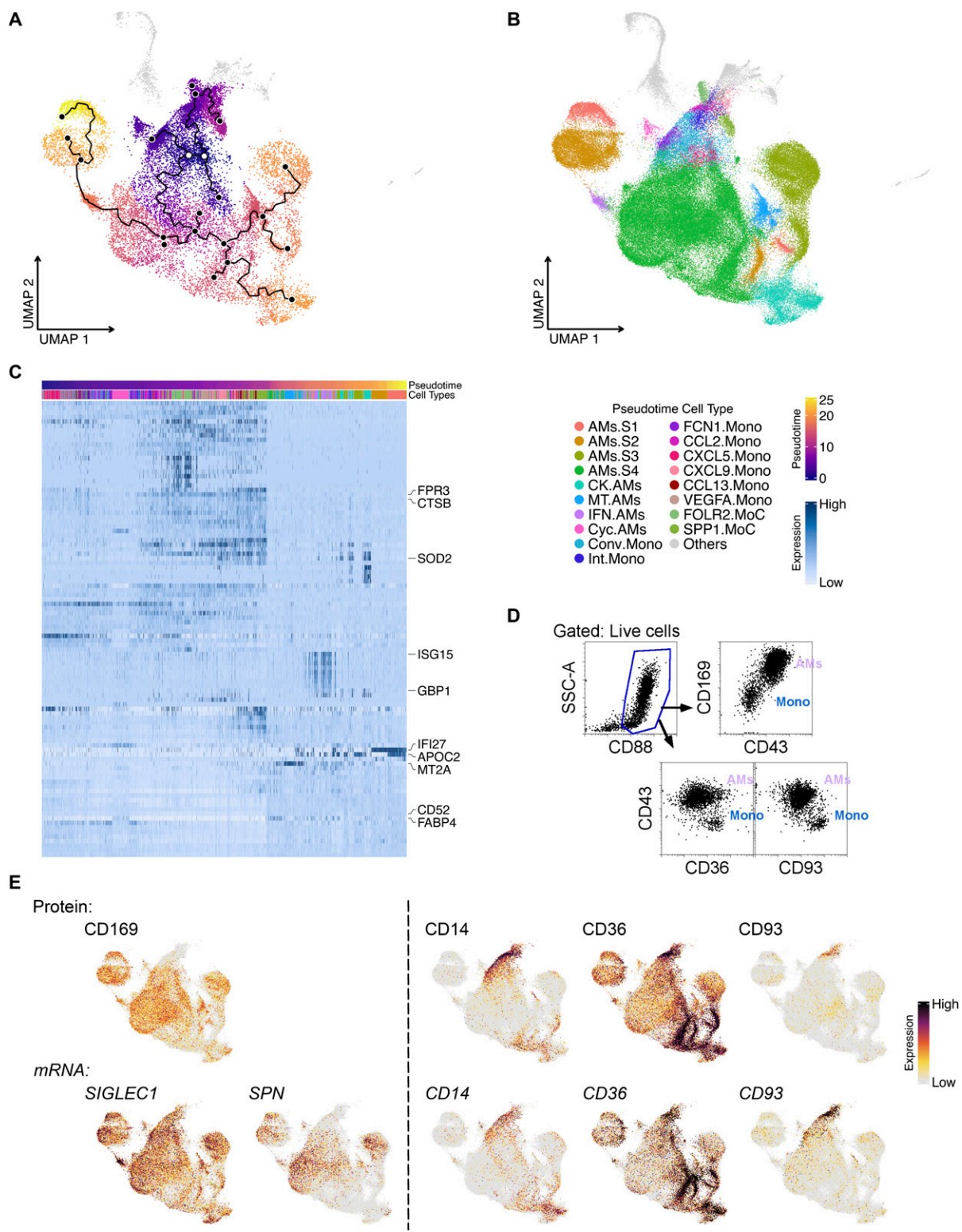

**Figure 6. Pseudotime analysis of monocytes and macrophages in BALF.**
**(A)** Pseudotime plot shows the evolvement of monocytes to macrophages with the root as white dots and the principal points as solid black dots. **(B)** UMAP shows distinct monocyte and macrophage clusters included for Pseudotime analysis. **(C)** Heat map shows the top 100 pseudotime differentially expressed genes across the monocyte-to-macrophage branch. Top 10 differentially expressed genes are labeled. **(D)** Flow cytometry plot illustrates the high expression of CD43 and CD169 on alveolar macrophages and CD93, CD36, and CD14 on monocytes. **(E)** Featured plots show representative gene expression and TotalSeq detection in alveolar macrophages and monocytes.

whether the four other non-supercluster AMs express all four combinations of *IFI27* and *APOC2* (Fig 7A). Indeed, we observed a quadrant distribution of *IFI27* and *APOC2* for Chemokine, Metal-lothionein, and Interferon AMs (Fig 7B). This suggests that non-supercluster AMs clusters are a part of the four AM superclusters. And the reason this was not observed in the analysis in Fig 2B is because of the overpowering expression of other functional genes in the non-supercluster AMs. Based on this observation, we hypothesize that discrete compartments in the lung dictate the expression of *IFI27* and *APOC2*, and within these compartments, a family of AMs containing all the AM subtypes, exist to ensure proper physiological functions (Fig 7C).

Subsequently, we examined whether AM superclusters occur in aggregated HC or CF, and in each individual sample. Unbiased analysis of each sample, or combined samples, demonstrates the formation of four unique AM superclusters (Fig S4). Thus, the robust recurrence of AM superclusters in each sample further validates our discovery and analysis.

**AM superclusters are conserved across different cohorts of publicly available datasets**

Last, we examined other AM datasets to ensure that our findings are not unique to our dataset. Six distinct datasets were selected based on the tissue, processing procedures, health status and cell

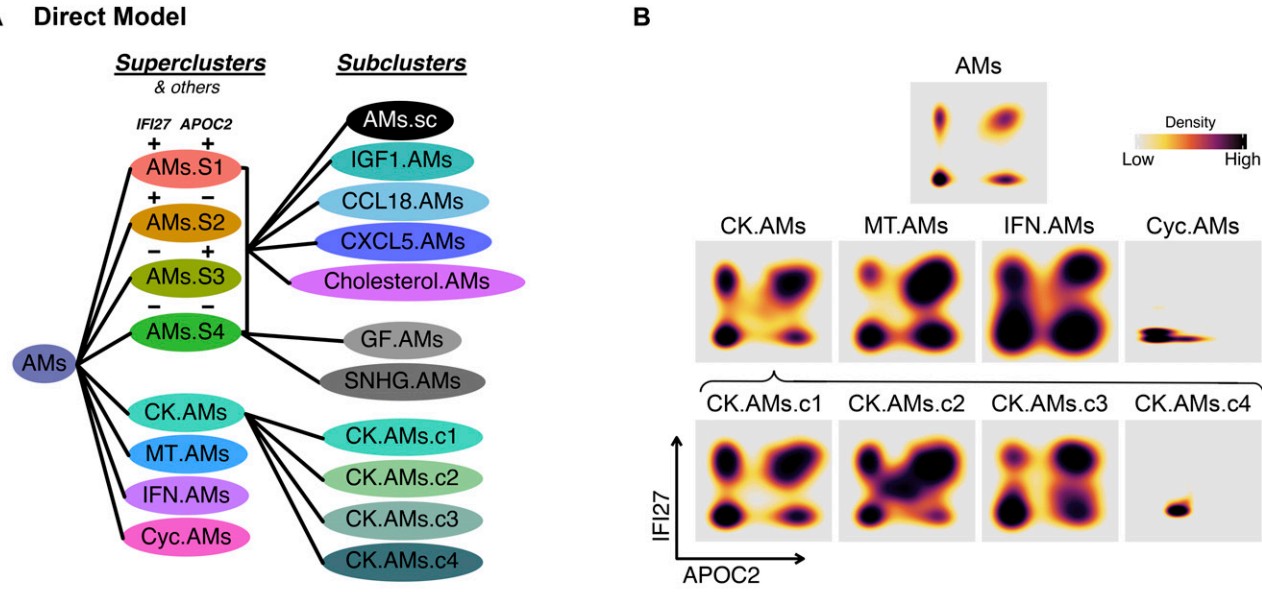

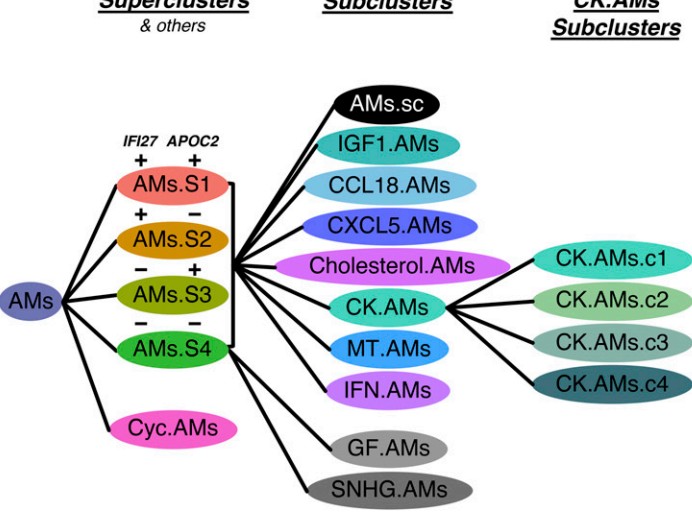

**Figure 7. Quadrant distribution of alveolar macrophage (AM) subtypes for the bivariate expression of AM subtypes for the of *IFI27* and *APOC2* and summary of proposed AM subtype classification.**
**(A)** AM subtype model supported by direct data. **(B)** Density plot showing the expression of *IFI27* and *APOC2* for chemokine, metallothionein, interferon, and cycling AMs. **(C)** Alternative AM subtype model proposed in the results section and discussion.

numbers (Fig 8A) (Vieira Braga et al, 2019; Liao et al, 2020; Schupp et al, 2020; Mayr et al, 2021; Mould et al, 2021; Dominguez Conde et al, 2022). Within the samples, only AMs were selected and re-clustered for UMAP visualization. Alignment of the top DEGs from AM clusters was performed (Fig 8B). All datasets contained the known AM subtypes that expressed chemokine, metallothionein, interferon and cycling genes. However, the superclusters defining *IFI27* and *APOC2* expression were only observed in the datasets of Liao et al (2020) and Dominguez Conde et al (2022). After accounting for sample size and sequencing depth, we hypothesize that the

difference between the presence of AM superclusters from one dataset versus another may be due to the processing procedures. Those that form superclusters need to express both *IFI27* and *APOC2* in the AMs, and only the samples that were freshly loaded expressed *APOC2*. It appears that freezing and thawing BAL or lung samples results in the loss of *APOC2* mRNA expression, explaining why superclusters could not be observed in these samples; thus, suggesting that there may be some genes sensitive to freeze/thaw procedures. Finally, using the Seurat "MapQuery" function, we projected datasets from other investigators onto our UMAP

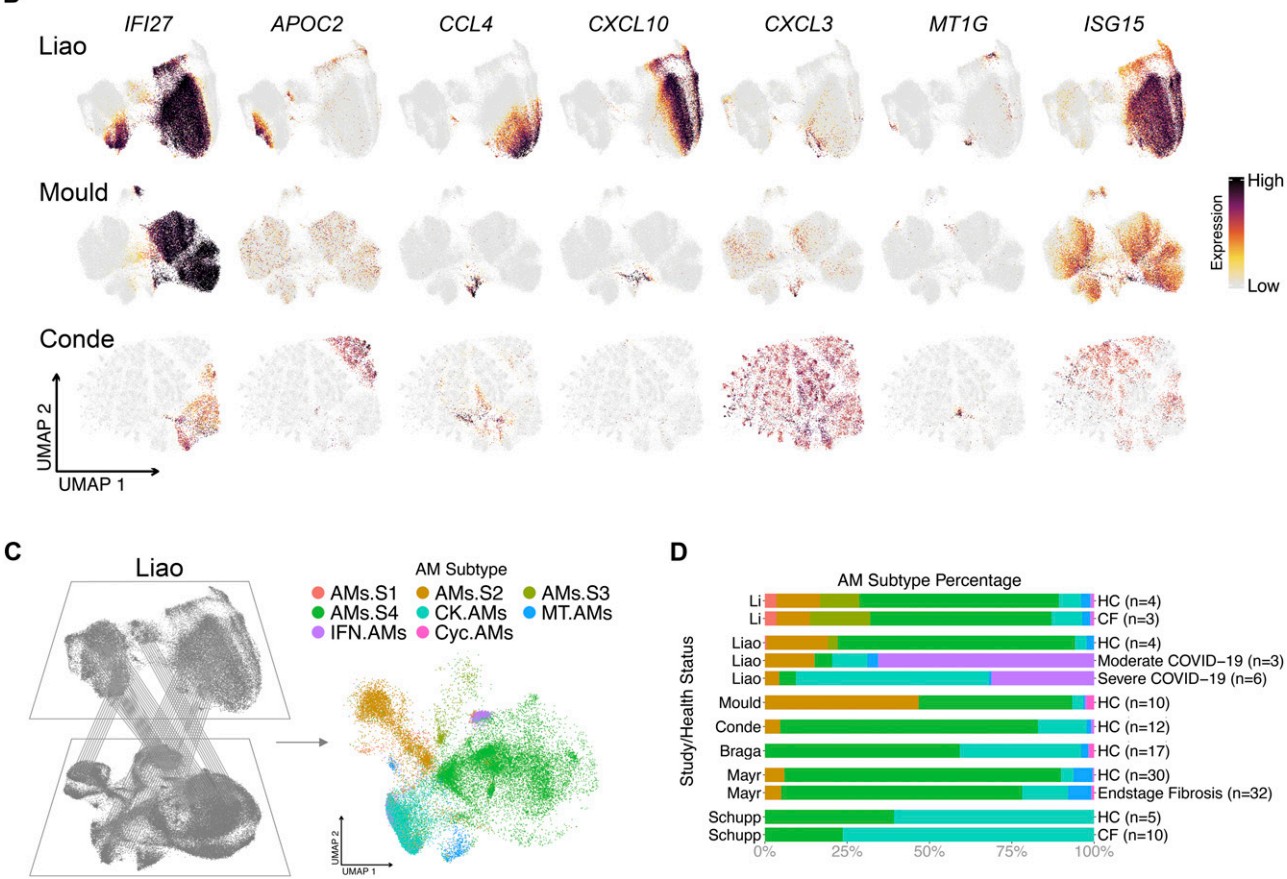

**Figure 8. Alveolar macrophage (AM) superclusters are present in other publicly available datasets.**
**(A)** AM summary table of publicly available datasets that were compared with our dataset. **(B)** Featured plots show the expression of seven selected AM cluster markers (*IFI27*, *APOC2*, *CCL4*, *CXCL10*, *CXCL3*, *MT1G*, and *ISG15*) in represented datasets. Only AMs were selected and re-clustered for UMAP visualization. **(C)** Schematic representation where reference mapping with Seurat package allows for the projection of publicly available datasets onto our AM supercluster-annotated dataset from Fig 2. **(D)** Percentage bar graph outlines the distribution of individual AM clusters for each health status in different datasets.

visualization from Figs 2–4, 8C and D, and S5, where the overall distribution was assessed. As anticipated, samples collected from the BAL and freshly processed displayed the best overlay (Liao et al, 2020). Finally, Liao et al, also had COVID-19–infected BAL samples that demonstrated significantly more CK.AMs and IFN.AMs present compared with healthy control samples (Liao et al, 2020), suggesting a functional shift in proinflammatory AM subtypes during SARS-CoV-2 infection.

## Discussion

AMs have multiple functional roles, including clearing surfactant components, cellular debris, inhaled particulates, and pathogens. Therefore, it is not surprising that in the airspace, there would be multiple AMs and monocyte subtypes with different specializations. ScRNA-seq technology has revolutionized our understanding of AMs, and we are only beginning to appreciate the depth and complexity of airway macrophages and monocytes in healthy and diseased lungs. Here, we used scRNA-seq to show that AMs form four superclusters based on the expression of two genes, *IFI27* and *APOC2*, whether integrated or independent. *IFI27* is a protein-coding gene regulated by interferon and estradiol (Rasmussen et al, 1993). It has been reported that IFI27 protein plays diverse roles in anti-tumor immunity, including suppressing the transcription of PD-L1 and promoting epithelial–mesenchymal transition (Li et al, 2015; Deng et al, 2020). In the lung, *IFI27* expression has also been suggested as a biomarker to differentiate influenza and bacterial infections (Tang et al, 2017). *APOC2* encodes a lipid-binding protein belonging to the apolipoprotein gene family, functioning as a co-factor to activate lipoprotein lipase for the hydrolyzation of tri-glycerides. The unique expression pattern of *IFI27* and *APOC2* generates four AM superclusters. Each supercluster contains subclusters defined by characteristic DEGs, including those for cytokines and chemokines. In addition to AM superclusters, we observed four other AM clusters. Of particular interest is the Chemokine AM cluster, which divides further into four subclusters. Each of these chemokine clusters expressed a combination of chemokines that selectively chemoattract different leukocytes into the environment. Strikingly, there appears to be a defined niche for each AM subtype given the tightly regulated frequency of each AM subtype across all samples.

Why do the superclusters and subclusters exist? Given the limited literature on the functionality of *IFI27* and *APOC2*, it is hard to speculate on the contribution of these two genes. In addition to the hypothesis that each AM supercluster exists in a discrete compartment of the lung, AM superclusters might regulate their environment in a tissue-specific niche and functionally specific manner. It would be scientifically and clinically interesting to know which CK.AM subclusters regulate the leukocyte environment in the airspace during and after infection with respiratory viruses like SARS-CoV-2.

Chemokines are chemoattractant cytokines, and this type of cytokine profiling resembles what has been observed for decades in T cell biology. Each T cell type, such as Th1 and Th2, selectively produces a characteristic set of cytokines. It is known that the differentiation of T cells is driven by distinct master regulators, transcription factors such as T-bet and Gata-3. Likewise, each AM

profiled in this study outlined a distinct set of transcription factors observed in our regulon analysis. Therefore, an important future goal would be to investigate which transcription factors regulate the individual Chemokine AM subtypes and beyond. Overall, the findings presented here open the door to identify new therapeutic targets for inflammatory lung disease while significantly enhancing our understanding of the complexity that exists with airspace macrophages.

Other immune cell types that are not normally thought to reside in airway space were captured in our BALF analysis: distinct DC types, monocytes and IMs. IMs are mainly investigated in mouse lungs, and multiple subtypes of IMs have been reported (Bedoret et al, 2009; Gibbings et al, 2017; Chakarov et al, 2019; Schyns et al, 2019; Ural et al, 2020). Among those subtypes, the well-described *FOLR2* positive IMs also exists in the human BAL cell population, suggesting homology across species. Another IM population denoted as SPP1.IMs because of the high expression of SPP1 (Leach et al, 2020). This IM subtype seems to be consistent with previously reported *SPP1/MERTK*-expressing macrophages in idiopathic pulmonary fibrosis (Morse et al, 2019). We also report several common markers for both human IM populations, *LGMN* and *MARCKS*, observed in a previous study (Reyfman et al, 2019).

This study was originally designed to investigate the cellular and transcriptional differences in cells extracted from BALF between a cohort of CF subjects with normal lung function and minimal lung inflammation compared with healthy controls to investigate the impact of aberrant CFTR function. Many lines of evidence point to impaired innate immune cell function in the CF lung (Gardai et al, 2005; Di et al, 2006; Murphy et al, 2010; Bessich et al, 2013; Paemka et al, 2017; Chen et al, 2018; Ruytinx et al, 2018; Hazlett et al, 2020); however, we observed some minor differences between CF and HC BALF cells (Complete list of genes: Tables S6–S9 and Fig S6), and it is unclear whether these differences are true across all CF patients. What is striking is that even though all three CF subjects had two copies of the F508del *CFTR* mutation, when the CF lung is in a healthy-like state, the macrophage, monocyte, IM, and DC heterogeneity is highly conserved, and its niche appropriately filled. All in all, the CF subjects in our study had preserved lung function and limited inflammation by design. Future studies could include subjects with more significant lung disease to determine the impact of inflammatory milieu on CF immune cell gene expression. In addition, comparisons of CF subjects with and without lung disease could help identify factors in airspace immune cells that predispose one to disease progression. Overall, several fundamental questions require further investigation, including the AM cluster location, function, transcription factor(s) that regulate their programming state, and whether similar macrophage superclusters exist in other organs or species.

## Materials and Methods

### Human subjects

Four healthy control (HC) subjects and three subjects with CF underwent research bronchoscopy as previously described (Hogan et

al, 2016; Aridgides et al, 2019). Briefly, after local anesthesia to the posterior pharynx and intravenous conscious sedation, a flexible fiberoptic bronchoscope was inserted transorally and passed through the vocal cords into the trachea. The bronchoscope was sequentially wedged into tertiary bronchi in the right upper lobe, right middle lobe, and right lower lobe, saline was instilled, and BALF was collected. An aliquot of 10 ml of BALF from the right middle lobe was used for the studies outlined here.

### Sample processing

Sample processing and library construction were performed immediately after lavage. 1X PBS containing 0.5 mM EDTA and 0.1% BSA was added to the BALF. Samples were filtered through 100-$\mu$m cell strainers and centrifugated at 300$g$ for 5 min at 4°C, and supernatants were discarded. TotalSeq antibodies were added at concentrations recommended by the manufacturer and stained for 30 min on ice (Stoeckius et al, 2017), details in online supplement. After staining, chilled HBSS supplemented with 0.5% BSA was used to wash the sample. Samples were again filtered through 100-$\mu$m cell strainers and spun as above. Supernatants were aspirated, and pellets were resuspended with HBSS plus 0.5% BSA at an approximate concentration of 7.5 × 10$^5$ cells/ml. Cell quality and viability were assessed with a Cellometer K2 (Nexcelom Bioscience). All samples had viability >80%. Single cells were then processed using the Chromium Next GEM Single Cell 3′ Platform (10X Genomics). Approximately 30,000 cells were loaded on each channel with an average recovery rate of 24,000 cells. Libraries were sequenced on NextSeq 500/550 (Illumina) with an average sequencing depth of 50,000 reads/cell.

### Data preparation

Raw sequencing reads were demultiplexed, mapped to the GRCh38 human reference genome, and gene expression matrices were generated using CellRanger v6.1 (10X Genomics). The following analyses were conducted in R 4.1 (R Core Team, 2014) and Python 3.6. Seurat package v4.0 was used for downstream data analyses (Hao et al, 2021), and figures were produced using the package ggplot2. Following a standard workflow, the gene expression matrix was filtered to discard cells with less than 200 genes, as well as genes that were expressed in less than three cells. Samples then underwent quality control to remove cells with either too many or too few expressed genes (average around 2,000 and 7,000) and cells with too many mtRNA (average around 10%), resulting in a total of 113,213 cells. Then, "SCTransform" was applied with the "glmGamPoi" method to normalize gene expression data (Stuart et al, 2019; Ahlmann-Eltze & Huber, 2021), and a CLR transformation was applied to normalize protein data within each cell. After individual preparation, all the samples were introduced into a combined Seurat object via "FindIntegrationAnchors" and "IntegrateData" functions (Stuart et al, 2019). Then scaled values of variable genes were then subject to principal component analysis for linear dimension reduction. A shared nearest neighbor network was created based on Euclidean distances between cells in multidimensional PC space (the first 50 PC were used) and a fixed number of neighbors per cell (30 neighbors). This

was used to generate a two-dimensional Uniform Manifold Approximation and Projection (UMAP) for visualization as a harmonized atlas to dissect the cell populations in BALF of healthy and CF donors.

### TotalSeq antibodies

TotalSeq antibodies were added at concentrations recommended by the manufacturer and stained for 30 min on ice (Stoeckius et al, 2017). CD88 (B1046), CD1c (B0160), CD1a (B0402), CD64 (B0162), CD14 (B0081), CD93 (B0446), CD197 (B0148), XCR1 (B0208), CD163 (B0358), CD36 (B0407), CD206 (B0205) HLA-DR (B0159), and CD169 (B0206) TotalSeq B (BioLegend). TotalSeq antibodies were added to the last four samples (HC3, HC4, CF2, and CF3).

### Differentially expressed genes

DEGs were calculated with "FindAllMarkers" function of Seurat in R 4.1 to study the different expression profiles in different cell types and up-regulated genes in HC or CF cohort (R Core Team, 2014; Hao et al, 2021). The "data" matrices of "SCT" assay were used, and the minimal log fold change was set to 0.25. Only genes that were detected in more than 25% of cells in either of the two populations were used to compute the DEGs with the Wilcoxon rank-sum test. Markers were identified as genes exhibiting significant up-regulation when compared against all other clusters and defined by having a Bonferroni-adjusted $P$-value < 0.05. The DEGs are ranked by the adjusted $P$-value to select the top DEGs for downstream analysis.

### Cell type identification

To identify cell types, "FindClusters" function with the Leiden algorithm with the various resolution from 0.5 to 2.0 in the Seurat package were used for clustering (Hao et al, 2021). "FindAllMarkers" function was then applied. The top DEGs of individual clusters were examined for well-studied marker genes across literature and the clusters were then annotated for the most likely identity. For re-clustered AMs and Mono subtypes, "FindClusters" was performed with different resolution until the right resolution is reached so that each cluster has unique gene expression pattern.

### Cross-sample cell type replicability

To compare cross-sample and cross-dataset transcriptomes, we performed MetaNeighbor (Crow et al, 2018) analysis between individual samples from our dataset and publicly available datasets addressing similar questions, (Schupp et al, 2020; Mould et al, 2021). Datasets were downloaded from UCSC Cell Browser (Kent et al, 2002) (https://cells.ucsc.edu/?ds=healthy-bal; https://cells.ucsc.edu/?ds=human-sputum) and gene expression omnibus (GEO; GSE151928; GSE145360). SummarizedExperiment objects were constructed using combined "data" matrices of SCT assays and the "variableGenes" function of MetaNeighbor package was used to identify the highly variable genes as input of similarity assessment (Crow et al, 2018). The cell–cell similarities were represented by the mean area under the receiver operator characteristic curve (AUROC) scores, keeping the original cell type from every dataset. The correlation matrices were

then passed to ComplexHeatmap package and clustered with "spearman" method (Gu et al, 2016). Heat map splits were applied to highlight the focus of the study.

### Gene ontology analysis

The gene set functional analysis was conducted with TopGO (Marini & Binder, 2019) (Fisher's exact test) to get a general overview of the more represented functional categories with the DEGs in each cluster. Background genes of gene enrichment assays were defined as all the detected genes in datasets and the biological process was selected as the target of ontology analysis. The DEGs of different cell types were used as the enrichment input. Top 5 GO terms ranked by *P*-value (Rantakari et al, 2016) for every cell type was selected and compared with other cell types. Overlaps of those GO terms were removed from the final visualization.

### Regulon analysis

All the cell types were downsampled to maximal 1,000 cells. TF activity was analyzed using SCENIC package of R 4.1 and pySCENIC of Python 3.8 for the cell types with downsampled count matrices as the input (Huynh-Thu et al, 2010; Aibar et al, 2017; Van de Sande et al, 2020). The regulons and TF activity (area under the curve) for each cell were calculated with motif collection version v9 as well as hg38__refseq r80__10kb_up_and_down_tss.mc9nr.featherda-tabases from the cisTarget (https://resources.aertslab.org/cistarget/). The transcription factor regulons prediction was performed using the pySCENIC default parameters (Huynh-Thu et al, 2010; Aibar et al, 2017; Van de Sande et al, 2020). The resulting AUC scores per each cell and adjacency matrices were used for downstream quantification of regulon specificity score. Top 5 transcription factor ranked by specificity score for every cell type was selected and compared with other cell types. Overlaps of those transcription factors were removed from the final visualization.

### Pseudotime analysis

The package Monocle3 was performed to analyze single-cell trajectories (Trapnell et al, 2014; Qiu et al, 2017a, 2017b; Cao et al, 2019). Pseudotime values were determined for all myeloid cells except DCs, which have a different ontogeny. All the cell types were downsampled to maximal 1,000 cells. Then Monocle3 was used to learn the sequence of gene expression changes each cell must go through as part of a dynamic biological process and then arrange each cell in the trajectory (Trapnell et al, 2014; Qiu et al, 2017a, 2017b; Cao et al, 2019). Two nodes of monocytes population were set as the start allowing the visualization of the differentiation process of monocytes to macrophages. To investigate the dynamic gene expression between monocytes and macrophages, DEGs over the pseudotime were calculated by the "graph_test" function. By ranking the *P*-value, top 100 pseudotime DEGs were identified. By further downsampling the cell types to maximal 100 cells, the expression levels of the top 100 pseudotime DEGs were then visualized and the top 10 were labeled.

### Cross-sample reference mapping

Publicly available AM/monocyte datasets were downloaded from UCSC Cell Browser (Kent et al, 2002) (https://cells.ucsc.edu/?ds=covid19-balf; https://cells.ucsc.edu/?ds=healthy-bal; https://cells.ucsc.edu/?ds=pan-immune+myeloid; https://cells.ucsc.edu/?ds=teichmann-asthma+lung-atlas+lung-atlas-others; https://cells.ucsc.edu/?ds=human-sputum), gene expression omnibus (GEO; GSE151928; GSE145360), and GitHub (https://github.com/theislab/2020_Mayr). To investigate the newly identified AM/monocyte subtypes in publicly available datasets, "FindTransfer-Anchors" and "TransferData" functions in the Seurat package were used to infer the cell types in the aforementioned query datasets (Stuart et al, 2019). The query cells were classified into different AM/monocyte subtypes based on their expression profile similarity with our dataset. Prediction scores were also returned to evaluate the reference prediction. Moreover, "MapQuery" function was applied to project the query datasets onto the our/reference UMAP structure for a more straightforward visualization (Stuart et al, 2019).

### Flow cytometry

BAL cells were stained with the following monoclonal Abs: anti-human CD45 BUV805 clone Hl30, anti-human CD14 V500, clone M$\phi$P9, anti-human HLA-DR APC-Cy7, clone L243, anti-human CD206 PerCp-Cy5.5, clone 15-2, anti-human CD36 APC, clone AC106, anti-human CD169 FITC, and anti-human CD93 PE-Cy7. The viability dye DAPI (#D9542; Sigma-Aldrich) was added immediately before each sample acquisition on a BD Symphony A3 analyzer (BD Biosciences). Data were analyzed using FlowJo (Tree Star). Antigen-specific antibodies and isotype controls were obtained from BioLegend, eBioscience (Ebioscience, AUT) and BD Biosciences.

## Data Availability

The sequencing raw data and processed data used in this article are available in Gene Expression Omnibus, accession code GSE193782. The processed data are also available for online visualization at https://ams-supercluster.cells.ucsc.edu (visualization for Figs 2–4 can be viewed with the "AMs Subset UMAP" layout).

### Study approval

All subjects provided written informed consent, and this study was approved by the Institutional Review Board of Dartmouth-Hitchcock Medical Center (protocol #22781).

## Supplementary Information

## Acknowledgements

Thank you to Dr. Tor Wager and William T King for critical feedback. This work was supported by National Institutes of Health grants R01 HL115334, R01 HL135001, and R35 HL155458 (CV Jakubzick). 10× Genomics and Illumina sequencing workflows were performed in collaboration with the Genomics Shared Resource and Single Cell Genomics Cores at Dartmouth, which receives support from NCI CCSG grant P30 CA023108, NIGMS P20 GM130454, and S10 OD025235.

## Author Contributions

X Li: conceptualization, resources, data curation, software, formal analysis, validation, investigation, visualization, methodology, and writing—original draft, review, and editing.

FW Kolling: resources, data curation, software, funding acquisition, methodology, and writing—review and editing.

D Aridgides: resources, methodology, and writing—review and editing.

D Mellinger: resources and methodology.

A Ashare: conceptualization, resources, data curation, supervision, funding acquisition, investigation, methodology, project administration, and writing—original draft, review, and editing.

CV Jakubzick: conceptualization, resources, data curation, supervision, funding acquisition, investigation, visualization, methodology, project administration, and writing—original draft, review, and editing.

## Conflict of Interest Statement

The authors declare that they have no conflict of interest.

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
