## [Reviewer comments · Life Science Alliance]

Life Science Alliance

ScRNA-seq Expression of IFI27 and APOC2 Identifies Four AM Superclusters in Healthy BALF

Xin Li, Fred Kolling, Daniel Aridgides, Diane Mellinger, Alix Ashare, and Claudia Jakubzick

DOI: <https://doi.org/10.26508/lsa.202201458>

Corresponding author(s): *Claudia Jakubzick, Dartmouth College and Alix Ashare, Geisel School of Medicine at Dartmouth*

Review Timeline:

Submission Date:	2022-03-20
Editorial Decision:	2022-04-22
Revision Received:	2022-05-25
Editorial Decision:	2022-06-16
Revision Received:	2022-06-20
Accepted:	2022-06-27

Transaction Report:

April 22, 2022

Re: Life Science Alliance manuscript #LSA-2022-01458-T

Dr. Claudia V. Jakubzick
Dartmouth College
Department of Microbiology and Immunology
626W Borwell
One Medical Center Drive
Lebanon, NH 3756

Dear Dr. Jakubzick,

Thank you for submitting your manuscript entitled "ScRNA-seq Expression of IFI27 and APOC2 Identifies Four Alveolar Macrophage Superclusters in Healthy BALF" to Life Science Alliance. The manuscript was assessed by expert reviewers, whose comments are appended to this letter. We invite you to submit a revised manuscript addressing the Reviewer comments.

Thank you for this interesting contribution to Life Science Alliance. We are looking forward to receiving your revised manuscript.

Sincerely,

B. MANUSCRIPT ORGANIZATION AND FORMATTING:

Reviewer #1 (Comments to the Authors (Required)):

In this study Li et al. perform an in-depth single-cell transcriptional analysis of alveolar macrophages (AM) from healthy volunteers and individuals with "mild" cystic fibrosis (CF). Their analysis reveals the existence of several distinct transcriptional programs and cell states, suggesting possible functional diversification and/or specialized niches for AM. While the findings presented in this paper are largely descriptive and confirmatory of similar published studies, this work does provide further insight into some of the transcriptional programs that differentiate unique clusters of AM in healthy individuals (particularly relating to chemokine profiles), and offers a useful resource for the field. However, given the descriptive nature of the study, this manuscript would benefit from (1) additional contextualization of the data in relation to AM biology, (2) comparison of the dataset to those of published datasets to establish whether the proposed AM superclusters are conserved across different cohorts, (3) further discussion of how the proposed framework for human AM classification could be useful/relevant to the field going forward.

Specific comments:

1. IFI27 and APOC2 are identified as markers for four overarching AM "superclusters" within which previously identified AM clusters fit. In the Discussion, the Authors posit these superclusters may be indicative of compartmentalization within the lung. As this is one of the major findings reported in the manuscript, further discussion is required to establish the relevance of the IFI27, APOC2 classification. Eg. Is there any evidence that these genes regulate specific functional programs in macrophages or tissue localization? Are there any other genes that are co-regulated with IFI27 and APOC2? If so, do these core gene sets reveal cluster-specific regulatory elements or pathways?

The pseudotime analysis suggests that there may be a temporal relationship between the superclusters with S1 and S2 most distant from monocytes. How does this relate to the compartmentalization of the superclusters?

2. Following on from above, have the Authors validated the IFI27, APOC2 classification at the protein level by flow cytometry analysis? Antibodies for both appear to be available (albeit polyclonal). This would be useful for the field as not all labs have access to scRNASeq technology.

3. Several other scRNASeq datasets of human BALF have been published recently and are referred to by the Authors. A cell-type comparison between the published studies and this one was presented in Fig1E but no further comparisons were made. An exploration of these published datasets in relation to the dataset in this study is required to establish whether the superclusters/subclusters identified are conserved across cohorts. It would be particularly interesting to compare the chemokine AM subclusters in datasets from disease contexts eg asthma (Li et al. 2021 JACI).

Minor comments:

1. Please define CF as cystic fibrosis in Abstract and first mention in the text.
2. Ref32 appears to be incorrectly referenced on pg6 in relation to previous AM cluster identification.
3. Panels 1C and 2C appear to be duplicates.
4. Fig3B is not clear to me. In the text the superclusters are reported to each contain IGF1, CCL18, CXCL5 and Cholesterol AM while s4 contains GF and SNHG AM, so why are the superclusters and subclusters plotted separately? Please clarify.
5. AMs.S4 is missing from panel 2D.
6. The DEG and Curated gene labels are incorrectly applied to Fig4D.

Reviewer #2 (Comments to the Authors (Required)):

This is a very interesting study that contains a wealth of information on the heterogeneity of AMs, IMs and monocytes in healthy human lung and very mild CF. The major finding is that the different subsets (or stable differentiation/specialization states) are

very conserved between patients, and that there is exquisite differentiation and much more heterogeneity than previously anticipated. This was an eye opener to me. Overall I greatly appreciated the amount of detail in this work, and this was an easy read.

Some suggestions for the authors :

- the CF patients were said to be free of inflammation. Please specify briefly in main text how this was addressed.
- the terms cluster and supercluster are already rapidly introduced in first 2 results section, but it would be nice to clarify what defined a supercluster or cluster : was this hierarchical clustering ? What defined a (super)cluster definition ? Were AM clusters identified only with the directed clustering or heterogeneity already seen in the initial full cell set ?
- please specify in main text how many AMs were used for the AM subclustering
- please specify in main text how many reads per cell are needed to obtain the degree of heterogeneity seen. In other words, Why have others missed this ?
- particularly for the CK Am subclusters some comments why the same percentage is seen in all CF patients and HC would be in place. Would this mean CF really changes nothing to the AMs ? Also the authors need to consider whether these are really subsets or merely activation states of the same population, maybe even induced by the stress of these cells taken out of their niche. I guess all of this was done really rapidly so mRNA upregulation induced by procedure seems unlikely. This could be part of the discussion. Finding the master TFs per state would solve much, but that's too much to ask for a first paper
- there is a paper on mouse AMs also suggesting they can be subdivided based on simple chemokine production. This could be cited. Similarly, the zmpfhregulin producing AMs have been described functionally in mice. Should be cited.

Reviewer #3 (Comments to the Authors (Required)):

In this very interesting manuscript, the authors are using single-cell RNA sequencing to characterize immune cell populations in the airways and alveoli, which were collected by performing bronchoalveolar lavages on healthy individuals and people with cystic fibrosis. They show that alveolar macrophages, the major immune cell population in uninflamed airways, can be clustered into four distinct families/superclusters based on the expression of IFI27 and APOC2 genes, among which further subclusters were identified. The authors initially also describe further superclusters, which they later in the article show are part of the novel four superclusters identified in this manuscript. This can be a little confusing and required repeated reading to follow. While it has been previously shown that AMs have different subsets, the findings in this manuscript give a very extensive picture of the landscape of AMs at steady state. Using pseudotime analysis the authors also point out that the different clusters are differentiating from both, monocytes and from the AMs.S4 supercluster, which are likely the main source of resident alveolar macrophages. This aligns perfectly with the literature. Additionally, the authors compare the described populations from healthy individuals with those of individuals with mild cystic fibrosis. The authors expand their analysis on monocyte populations, which they also show to be clustering into a total of 8 clusters. Overall, albeit descriptive, this is a very robust and relevant study, and strengthens the foundation for future research and potential therapeutical targets in the field of lung (immune) diseases.

Points:

- 1) It is difficult to appreciate the "strong correlation" in Figure 1G between the authors dataset and the dataset from Schupp et al. - while I agree that the cell populations are (for the most part) correlating, there seems to be a discrepancy in monocytes and AM (total). I would have loved to see the differences between these two datasets closer analyzed, especially in the Monocyte and AM (total) populations, and whether the same clusters would still be maintained among the samples from individuals with more severe lung damage. I believe such analysis could strengthen the manuscript further, considering that the samples in the Schupp et al. cohort had on average a (much) worse lung function, thus might give insights as to which populations are dysregulated compared to a healthier state. This comparison is mentioned by the authors as potential future study, but might be available to assess already.
- 2) In Figure 7B there is a black path leading to no subcluster. Figure 7C is not mentioned in the Results, the figure legend says that it relates to the discussion, but I did not find any reference to it. I believe the manuscript could benefit from a more detailed discussion about the final super/subclustering categories to make the reader understand on the initial read why some of the "superclusters" could/should be categorized as subclusters. I personally prefer 7C as a clustering model, as CK, MT and IFN AMs are also subcategorizing into all 4 IFI27/APOC2 superclusters, while Cyc.AMs seem to be part of AMs.S4 (which is in line with the Pseudotime analysis). Furthermore, it seems to me that AMs.S4 are the predominant population in accordance with Fig. 2E and F (and probably the corrected Figure 2C), this could also be pointed out in the Results/Discussion.
- 3) It is interesting that the authors find interstitial macrophages (and dendritic cells) in the airways, which have an immune regulatory phenotype. In accordance with Figure 5 (A and B) it seems like they are mostly represented in CF individuals (FOLR2.IMs mostly in CF1 and SPP1.IMs in CF2). Here I am just curious whether the CF1 and CF2 samples are by any chance from CF individuals with positive PA cultures?
- 4) It is a little bit concerning that no neutrophils were found/shown in the analysis, especially considering that two samples were from individuals with positive PA cultures. Were neutrophils absent in all samples, or were they somehow excluded from the analysis?

5) I would appreciate some information about APOC2 and IFI27 (at least the full gene names), as these are used to distinguish the novel superclusters.

6) I believe that Figure 1C and 2C are identical, please exchange Figure 2C with the correct graph.

7) Figure 5 only has A-E labels, but figure legend goes until G (Are Figures 5F and 5G missing, or is the Figure legend too extensive?)

8) Figure S3 needs a legend for the color code (at least in the figure legend).

Corrections for the text:

Page 3: The third paragraph's last sentence is missing a word or some other grammatical correction (... to be characterized in diseased lungs?)

Page 8 Title of second paragraph: "selective" should be used as adverb (selectively secreted)

Page 12 the second sentence of the second paragraph reads odd, maybe exchange "...due to many lines of evidence..." into "...as many lines of evidence..."

Point-by-point response

We thank all the reviewers and Editor for their thorough and thoughtful evaluation of our manuscript, resulting in an improved version and two new Figures, Figure 8 and Supplementary Figure 6, which compares our dataset with six other publicly available datasets.

We addressed the comments of all three reviewers below.

Reviewer #1

R1 In this study Li et al. perform an in-depth single-cell transcriptional analysis of alveolar macrophages (AM) from healthy volunteers and individuals with "mild" cystic fibrosis (CF). Their analysis reveals the existence of several distinct transcriptional programs and cell states, suggesting possible functional diversification and/or specialized niches for AM. While the findings presented in this paper are largely descriptive and confirmatory of similar published studies, this work does provide further insight into some of the transcriptional programs that differentiate unique clusters of AM in healthy individuals (particularly relating to chemokine profiles), and offers a useful resource for the field. However, given the descriptive nature of the study, this manuscript would benefit from (1) additional contextualization of the data in relation to AM biology, (2) comparison of the dataset to those of published datasets to establish whether the proposed AM superclusters are conserved across different cohorts, (3) further discussion of how the proposed framework for human AM classification could be useful/relevant to the field going forward.

C1a. We have added more information for the potential meaning of our data in the discussion. We created a new Figure and Supplementary Figure, Figures 8 and S6 to compare our dataset with six other publicly available datasets that were selected based on cellular extraction (i.e., contain healthy and diseased airspace leukocytes) and processing procedures. We found our defined AM cell types in all the datasets. However, not all the datasets had all our cell types. As for AM superclusters we found that only samples that were loaded fresh on a sequencing platform conserved the expression of APOC2, which is required to form AM superclusters. Thus, it seems like freeze-thaw procedures of samples results in the loss of APOC2 mRNA.

Specific comments:

1. IFI27 and APOC2 are identified as markers for four overarching AM "superclusters" within which previously identified AM clusters fit. In the Discussion, the Authors posit these superclusters may be indicative of compartmentalization within the lung. As this is one of the major findings reported in the manuscript, further discussion is required to establish the relevance of the IFI27, APOC2 classification. Eg. Is there any evidence that these genes regulate specific functional programs in macrophages or tissue localization? Are there any other genes that are co-regulated with IFI27 and APOC2? If

so, do these core gene sets reveal cluster-specific regulatory elements or pathways?

The pseudotime analysis suggests that there may be a temporal relationship between the superclusters with S1 and S2 most distant from monocytes. How does this relate to the compartmentalization of the superclusters?

C1b.

To check the expression of genes that coregulated with either IFI27 or APOC2, we first defined the top 50 target genes using COXPRESdb.ver 8.0. These targets were analyzed using expression heatmaps to view expression levels in different AM clusters. As anticipated by DEG analysis, none of the top 50 potentially coexpressed target genes show similar expression patterns in AM supercluster expression either IFI27 or APOC2.

Data presented here for reviewer:

Based on pseudotime analysis S4 appears to be the common origin for S1, S2 and S3. S3 is almost as equally distant as S1 and S2 to monocytes, suggesting that they all derive from S4 and upregulate IFI27, APOC2 or both.

R2. Following on from above, have the Authors validated the IFI27, APOC2 classification at the protein level by flow cytometry analysis? Antibodies for both appear to be available (albeit polyclonal). This would be useful for the field as not all labs have access to scRNASeq technology.

C2. We purchased IFI27 and APOC2 (IFI27 Antibody, Rabbit, Polyclonal, Novus USA NB300-709 and Apolipoprotein C-II/ApoC2 Antibody, goat, Polyclonal, Novus USA, NB400-155) and several secondary antibodies. We stained fresh human BAL and human lung paraffin-embedded sections. Unfortunately, the antibodies appeared to be non-specific. All cells shifted positive by flow, including cells that were not AMs. Regardless of the combination of secondaries we used. In the IHCs AMs were stained double positive, supporting the observation we made in flow, that the Abs were binding in non-specifically.

R3. Several other scRNASeq datasets of human BALF have been published recently and are referred by the Authors. A cell-type comparison between the published studies and this one was presented in Fig1E but no further comparisons were made. An exploration of these published datasets in relation to the dataset in this study is required to establish whether the superclusters/subclusters identified are conserved across cohorts. It would be particularly interesting to compare the chemokine AM subclusters in datasets from disease contexts eg asthma (Li et al. 2021 JACI).

C3. Thank you for this comment. We created a new figure and supplementary figure, Figure 8 and SupFig6 to compare our datasets with other publicly available datasets, as described in C1. We contacted both corresponding authors in this manuscript, Li et al. 2021 JACI, neither of them replied to our request for a link to their dataset since their data is not publicly available.

Minor comments:

1. Please define CF as cystic fibrosis in Abstract and first mention in the text.
--Done
2. Ref32 appears to be incorrectly referenced on pg6 in relation to previous AM cluster identification.
-- Fixed
3. Panels 1C and 2C appear to be duplicates.
--Thank you for catching this- fixed
4. Fig3B is not clear to me. In the text the superclusters are reported to each contain IGF1, CCL18, CXCL5 and Cholesterol AM while s4 contains GF and SNHG AM, so why are the superclusters and subclusters plotted separately? Please clarify.
--Clarified
5. AMs.S4 is missing from panel 2D.
--This is because the duplicates are removed so all the DEGs in AMs.S4 are shared with AMs.S1,2 and 3. The dot plots do not show duplicate genes, we have added this information in the figure legend.
6. The DEG and Curated gene labels are incorrectly applied to Fig4D.
--Thank you for seeing this. Corrected.

Reviewer #2 (Comments to the Authors (Required)):

This is a very interesting study that contains a wealth of information on the heterogeneity of AMs, IMs and monocytes in healthy human lung and very mild CF. The

major finding is that the different subsets (or stable differentiation/specialization states) are very conserved between patients, and that there is exquisite differentiation and much more heterogeneity than previously anticipated. This was an eye opener to me. Overall I greatly appreciated the amount of detail in this work, and this was an easy read.

Some suggestions for the authors :

R1. -the CF patients were said to be free of inflammation. Please specify briefly in main text how this was addressed.

C1. We have now added this information in the results section and a Table in the supplement.

R2. -the terms cluster and supercluster are already rapidly introduced in first 2 results section, but it would be nice to clarify what defined a supercluster or cluster : was this hierarchical clustering ? What defined a (super)cluster definition ?

C2. We added several sentences to clarify the definition of supercluster and clusters in the results section.

R3. -please specify in main text how many AMs were used for the AM subclustering

C3. This is now in Figure 1A. Thank you for this suggestion.

R4. -please specify in main text how many reads per cell are needed to obtain the degree of heterogeneity seen.

C4. We have added the sequencing depth in the methods section.

R5. -particularly for the CK Am subclusters some comments why the same percentage is seen in all CF patients and HC would be in place. Would this mean CF really changes nothing to the AMs ?

C5. We have added a few sentences in the discussion hypothesizing why mild-CF subjects in our cohort have similar AM percentages as HC subjects.

R6. Also the authors need to consider whether these are really subsets or merely activation states of the same population, maybe even induced by the stress of these cells taken out of their niche. I guess all of this was done really rapidly so mRNA upregulation induced by procedure seems unlikely. This could be part of the discussion.

C6. Indeed this was performed rapidly and the conservation across all seven dataset would suggest little alteration in mRNA expression due to processing. Our dataset suggests functional difference more than activation states. And even if they are different activation states, the data would suggest they perform different functions such as the

MT AMs, all its top DEGs are metallothioneins; whereas no other AM subtypes has expression of metallothioneins. This supports the concept that these AMs display differences in function more than activation states.

R7. -there is a paper on mouse AMs also suggesting they can be subdivided based on simple chemokine production. This could be cited. Similarly, the amphiregulin producing AMs have been described functionally in mice. Should be cited.

C7. Papers cited

Reviewer #3 (Comments to the Authors (Required)):

In this very interesting manuscript, the authors are using single-cell RNA sequencing to characterize immune cell populations in the airways and alveoli, which were collected by performing bronchoalveolar lavages on healthy individuals and people with cystic fibrosis. They show that alveolar macrophages, the major immune cell population in uninflamed airways, can be clustered into four distinct families/superclusters based on the expression of IFI27 and APOC2 genes, among which further subclusters were identified. The authors initially also describe further superclusters, which they later in the article show are part of the novel four superclusters identified in this manuscript. This can be a little confusing and required repeated reading to follow. While it has been previously shown that AMs have different subsets, the findings in this manuscript give a very extensive picture of the landscape of AMs at steady state. Using pseudotime analysis the authors also point out that the different clusters are differentiating from both, monocytes and from the AMs.S4 supercluster, which are likely the main source of resident alveolar macrophages. This aligns perfectly with the literature. Additionally, the authors compare the described populations from healthy individuals with those of individuals with mild cystic fibrosis. The authors expand their analysis on monocyte populations, which they also show to be clustering into a total of 8 clusters. Overall, albeit descriptive, this is a very robust and relevant study, and strengthens the foundation for future research and potential therapeutical targets in the field of lung (immune) diseases.

Points:

R1. It is difficult to appreciate the "strong correlation" in Figure 1G between the authors dataset and the dataset from Schupp et al. - while I agree that the cell populations are (for the most part) correlating, there seems to be a discrepancy in monocytes and AM (total). I would have loved to see the differences between these two datasets closer analyzed, especially in the Monocyte and AM (total) populations, and whether the same clusters would still be maintained among the samples from individuals with more severe lung damage. I believe such analysis could strengthen the manuscript further, considering that the samples in the Schupp et al. cohort had on average a (much) worse lung function, thus might give insights as to which populations are dysregulated

compared to a healthier state. This comparison is mentioned by the authors as potential future study, but might be available to assess already.

C1. Thank you for suggesting this. We have now performed this analysis. One thing to keep in mind is that Schupp dataset is freeze-thawed sputum, unlike our which is BAL loaded fresh on the 10X platform. Both the sample and processing are different. As shown in Fig 8 and Supplementary figure 6 we analyzed the cell type percentage in the sputum paper, where they have more AMs.S4 and FCN1 monocytes. This is interesting because it suggests that with severe CF there are niches that require filling and why AMs.S4 is most likely expanding along with the recruitment of monocytes.

R2. In Figure 7B there is a black path leading to no subcluster. Figure 7C is not mentioned in the Results, the figure legend says that it relates to the discussion, but I did not find any reference to it. I believe the manuscript could benefit from a more detailed discussion about the final super/subclustering categories to make the reader understand on the initial read why some of the "superclusters" could/should be categorized as subclusters. I personally prefer 7C as a clustering model, as CK, MT and IFN AMs are also subcategorizing into all 4 IFI27/APOC2 superclusters, while Cyc.AMs seem to be part of AMs.S4 (which is in line with the Pseudotime analysis). Furthermore, it seems to me that AMs.S4 are the predominant population in accordance with Fig. 2E and F (and probably the corrected Figure 2C), this could also be pointed out in the Results/Discussion.

C2. Thank you for these suggestions. We have incorporated these comments into our result and discussion section. 7B and 7C are fixed.

3) It is interesting that the authors find interstitial macrophages (and dendritic cells) in the airways, which have an immune regulatory phenotype. In accordance with Figure 5 (A and B) it seems like they are mostly represented in CF individuals (FOLR2.IMs mostly in CF1 and SPP1.IMs in CF2). Here I am just curious whether the CF1 and CF2 samples are by any chance from CF individuals with positive PA cultures?

C3. Thank you for making this observation. Once the reviewer pointed this out, we were also curious. CF1 was positive for PA, CF2 was positive for SA and CF3 was positive for PA.

R4) It is a little bit concerning that no neutrophils were found/shown in the analysis, especially considering that two samples were from individuals with positive PA cultures. Were neutrophils absent in all samples, or were they somehow excluded from the analysis?

C4. We did not see many neutrophils by flow cytometry above those of HC. Thus suggesting that even though bacteria was detected, it must have been a very mild infection. Moreover, AMs overwhelm all other cell types so when using the 10X platform, the most abundant cells tend to be acquired. To note: we did confirm in our CF

samples the double deletion of CFTR using our raw sequencing data in the epithelial cells.

5) I would appreciate some information about APOC2 and IFI27 (at least the full gene names), as these are used to distinguish the novel superclusters.

C5- This is now incorporated in the discussion.

6) I believe that Figure 1C and 2C are identical, please exchange Figure 2C with the correct graph.

C6. Thank you, corrected.

7) Figure 5 only has A-E labels, but figure legend goes until G (Are Figures 5F and 5G missing, or is the Figure legend too extensive?)

C7. Corrected

8) Figure S3 needs a legend for the color code (at least in the figure legend).

Corrections for the text:

Page 3: The third paragraph's last sentence is missing a word or some other grammatical correction (... to be characterized in diseased lungs?)

Page 8 Title of second paragraph: "selective" should be used as adverb (selectively secreted)

Page 12 the second sentence of the second paragraph reads odd, maybe exchange "...due to many lines of evidence..." into "...as many lines of evidence..."

C8. All fixed

June 16, 2022

RE: Life Science Alliance Manuscript #LSA-2022-01458-TR

Prof. Claudia Jakubzick
Dartmouth College
Department of Microbiology and Immunology
626W Borwell
One Medical Center Drive
Lebanon, NH 03756

Dear Dr. Jakubzick,

Thank you for submitting your revised manuscript entitled "ScRNA-seq Expression of IFI27 and APOC2 Identifies Four AM Superclusters in Healthy BALF". We would be happy to publish your paper in Life Science Alliance pending final revisions necessary to meet our formatting guidelines.

- please add ORCID ID for secondary corresponding author-they should have been sent instructions on how to do so
- please add the Twitter handle of your host institute/organization as well as your own or/and one of the authors in our system
- please use the [10 author names, et al.] format in your references (i.e. limit the author names to the first 10)
- please upload your graphical abstract as a separate file with the file designation 'Graphical Abstract'
- please upload your table files as separate files or include them in the doc file of your manuscript
- please add a Figure/Table Callout for figure 4A, Figure S2B, and Tables S2-S9 to your main manuscript text

A. FINAL FILES:

B. MANUSCRIPT ORGANIZATION AND FORMATTING:

Sincerely,

Reviewer #1 (Comments to the Authors (Required)):

The Authors have satisfactorily addressed all my comments.

Reviewer #2 (Comments to the Authors (Required)):

Authors have satisfactorily addressed my concerns. This is a nice study.

Reviewer #3 (Comments to the Authors (Required)):

The authors have provided ample responses to the previously raised concerns and have revised the manuscript appropriately. Especially the major concern about the comparability between the authors' and other datasets on alveolar macrophages as well as the differences have been addressed in the results and discussion with a new figure and supplementary figures. I have no further points to raise about the revised manuscript and believe that in its current form will provide a good basis to investigate disease-affected or disease-driving subpopulations of alveolar macrophages in the fields of cystic fibrosis, but also other inflammatory lung diseases.

June 27, 2022

RE: Life Science Alliance Manuscript #LSA-2022-01458-TRR

Prof. Claudia Jakubzick
Dartmouth College
Department of Microbiology and Immunology
626W Borwell
One Medical Center Drive
Lebanon, NH 03756

Dear Dr. Jakubzick,

Thank you for submitting your Research Article entitled "ScRNA-seq Expression of IFI27 and APOC2 Identifies Four AM Superclusters in Healthy BALF". It is a pleasure to let you know that your manuscript is now accepted for publication in Life Science Alliance. Congratulations on this interesting work.

DISTRIBUTION OF MATERIALS:

Again, congratulations on a very nice paper. I hope you found the review process to be constructive and are pleased with how the manuscript was handled editorially. We look forward to future exciting submissions from your lab.

Sincerely,
